# Elevating Augmentation:
# Boosting Performance via Sub-Model Training

## Abstract

Image classification has improved with the development of training techniques. However, these techniques often require careful parameter tuning to balance the strength of regularization, limiting their potential benefits. In this paper, we propose a novel way to use regularization called Augmenting Sub-model (AugSub). AugSub consists of two models: the main model and the sub-model. While the main model employs conventional training recipes, the sub-model leverages the benefit of additional regularization. AugSub achieves this by mitigating adverse effects through a relaxed loss function similar to self-distillation loss. We demonstrate the effectiveness of AugSub with three drop techniques: dropout, drop-path, and random masking. Our analysis shows that all AugSub improves performance, with the training loss converging even faster than regular training. Among the three, AugMask is identified as the most practical method due to its performance and cost efficiency. We further validate AugMask across diverse training recipes, including DeiT-III, ResNet, MAE fine-tuning, and Swin Transformer. The results show that AugMask consistently provides significant performance gain. AugSub provides a practical and effective solution for introducing additional regularization under various training recipes. The code will be available publicly.

## 1 Introduction

As deep neural networks scale up, addressing issues such as overfitting and improving generalization performance becomes essential. To solve this, various data augmentation and regularizations have been developed. Starting from the traditional techniques like weight-decay and dropout (Srivastava et al., 2014), modern approaches such as image-mixing augmentations (e.g., Mixup (Zhang et al., 2017) and CutMix (Yun et al., 2019)), mixture of data augmentation (e.g., RandAugment (Cubuk et al., 2020) and AutoAugment (Cubuk et al., 2019)), and drop-based techniques (e.g., Drop-path (Fan et al., 2019) and RandomErase (Zhong et al., 2020)) have been widely used to improve the performance.

These regularizations and augmentations usually improve generalization performance, making training difficult and hindering the deep models from converging with low training loss. In other words, the training techniques often underfit a model to the training data with degraded performance. Thus, practitioners and researchers have empirically found appropriate combinations and intensities of these techniques (Wightman et al., 2021; Touvron et al., 2021a; 2022a), a concept referred to as "training recipes". The importance of such training recipe becomes even more significant in Vision Transformer (ViT) (Dosovitskiy et al., 2020) architecture. The recipes of DeiT (Touvron et al., 2021a) and DeiT-III (Touvron et al., 2022a) are considered de facto for training ViTs.

A significant role of training recipes (Wightman et al., 2021; Touvron et al., 2022a) is to find optimal hyperparameters for training techniques, so modifying the recipes may lead to unstable training or degraded performance. This makes it challenging for users to increase the intensity of regularization or introduce a new training technique.

Our research goal is to achieve further performance improvements with additional regularization while maintaining the stability of existing training recipes. To this end, we introduce a training framework using a "sub-model" alongside the main model; throughout this paper, we use the term "sub-model" to describe a partial model extracted from the "main model", in which some trainable weights do not engage in training. The main model uses standard training recipes (Wightman et al., 2021; Touvron et al., 2022a). On the other hand, the sub-model utilizes additional regularization.

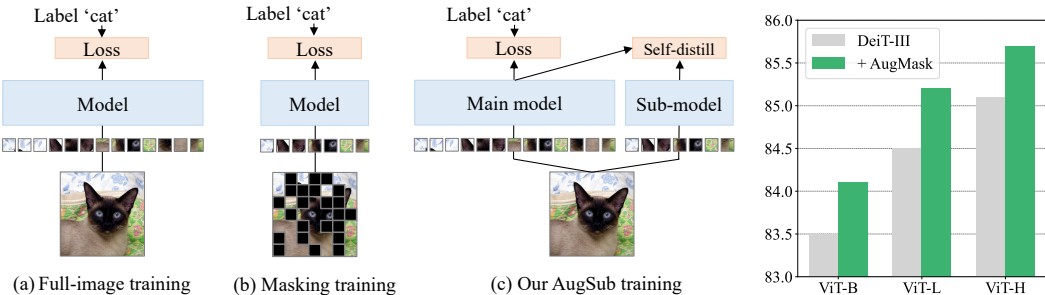

Figure 1: **Overview of our Augmenting Sub-model (AugSub).** (a) original supervised training; (b) conventional drop-based technique (random masking). It is applied to the main model, which degrades performance; (c) our proposed **AugSub** training, which separates the drop-based technique from the main model using the sub-model and employs a relaxed loss based on self-distillation. It achieves significant improvements from the state-of-the-art ViT training recipe (Touvron et al., 2022a).

We name our method as Augmenting Sub-model (AugSub). In the example of Figure 1, the desired additional regularization is random masking (as done in MAE (He et al., 2022)). As in Figure 1 (b), applying the random masking to the main model may lead to degraded performance. In contrast, as in Figure 1 (c), AugSub utilizes the sub-model for random masking, and the sub-model receives a training signal from the main model similar to the self-distillation (Zhu et al., 2018; Zhang et al., 2019; Phuong & Lampert, 2019). While the random masking technique amplifies the difficulty of the training process, this is counterbalanced by self-distillation loss since the outputs of the main model are relaxed and easier objective than the ground-truth label.

In summary, AugSub applies an additional regularization separated from the main model, utilizing a relaxed loss form. As a result, AugSub enables any additional regularization without disrupting the convergence of original train loss; we employ three strong in-network drop-based options to show the applicability: dropout (Srivastava et al., 2014), drop-path (Huang et al., 2016; Fan et al., 2019), and input masking (He et al., 2022; Bao et al., 2021). Corresponding to each respective regularization strategy, we denote them AugDrop, AugPath, and AugMask.

We extensively validate the performance of three AugSub methods. First, we analyze AugSub using 100 epochs training on ImageNet (Deng et al., 2009). Without AugSub, loss convergence speed and corresponding accuracy are significantly degraded when additional regularization is applied. Conversely, AugSub successfully mitigates potential harmful effects from additional regularization, leading to a network training process that is even more efficient than standard training procedures. Among the three variants of AugSub, AugMask notably exhibits a significant enhancement in performance. Thus, we expand experiments to regular training in ImageNet (Deng et al., 2009) focus on AugMask. AugMask is applied on various supervised learning cases including DeiT-III (Touvron et al., 2022a), ResNet-RSB (Wightman et al., 2021), MAE finetuning (He et al., 2022), and Swin transformer (Liu et al., 2021). AugMask demonstrates remarkable performance improvement in all benchmarks. We argue that AugMask can be regarded as a significant advancement as a novel way to utilize regularization for visual recognition.

## 2 RELATED WORK

**Training recipe** has been considered an important ingredient in building a high-performance network. He et al. (He et al., 2019) demonstrate that the training recipe significantly influences the network performance. RSB (Wightman et al., 2021) is a representative and high-performance recipe for ResNet. With the emergence of ViT (Dosovitskiy et al., 2020), the training recipe for ViT has gained the attention of the field. DeiT (Touvron et al., 2021a) shows that ViT can be trained to strong performance with only ImageNet-1k (Deng et al., 2009). DeiT-III (Touvron et al., 2022a) is an improved version of DeiT, which applies findings from RSB to DeiT instead of distillation from CNN teacher. It is challenging to implement stronger or additional regularization in existing training recipes. To address this issue, we propose AugSub approach employing sub-models.

CoSub (Touvron et al., 2022b) introduces a similar concept to ours, utilizing sub-models. However, the objective of the sub-model differs significantly: while AugSub aims to stabilize training through

additional regularization, CoSub aims to train the sub-models in a collaborative manner (Zhang et al., 2018). We regard AugSub as a more generalized framework since CoSub only considers the drop-path method to employ sub-models, whereas AugSub can cover a variety of drop-based techniques.

**Self-distillation** utilizes supervision from a network itself instead of using a teacher. ONE (Zhu et al., 2018) uses a multi-branch ensemble to build superior output for the network and distill ensemble outputs as supervision for each branch. Some studies (Zhang et al., 2019; Phuong & Lampert, 2019) utilize the early-exit network for self-distillation. Those studies improve performance by using a full network as a teacher and an early-exit network as a student. MaskedKD (Son et al., 2023) utilizes masking to reduce computation for knowledge distillation. From a self-distillation perspective, AugSub presents a new insight to construct the student model (i.e., sub-model) from the teacher model (i.e., main model) utilizing drop-based techniques.

**Self-supervised learning** shares components with AugSub. Previous works on contrastive learning incorporate two models with self-distillation loss (Chen & He, 2021; Grill et al., 2020). Want et al. (Wang et al., 2022) introduce a double tower with weak and strong augmentation for each model. In masked image models, supervised MAE (Liang et al., 2022) introduces additional supervised learning tasks to the MAE framework and accelerates MAE. Those studies partially share the fundamental concept with AugSub and gave us the motivation for AugSub. However, the proposed AugSub is significantly different from self-supervised learning approaches.

## 3 METHOD

We propose our method Augmenting Sub-model (AugSub) with formulation and pseudo-code in Section 3.1. Next, we introduce three variants of AugSub: AugDrop, AugPath, and AugMask in Section 3.2. Section 3.3 presents analyses of AugSub with loss convergence, accuracy, and gradient.

### 3.1 AUGMENTING SUB-MODEL (AUGSUB)

The cross-entropy loss with the softmax $\sigma(\mathbf{z}) = e^{z_i} / \sum_j e^{z_j}$ for images $\mathbf{x}_i$ and one-hot labels $\mathbf{y}_i (i \in [1, 2, ..., N])$ in a mini-batch with size $N$ is

$$-\frac{1}{N} \sum_i^N \mathbf{y}_i \log\left(\sigma(f_\theta(\mathbf{x}_i|p_{\text{drop}} = 0))\right), \tag{1}$$

where $f_\theta$ represents the network used for training. $p_{\text{drop}}$ means drop probability of network. Since the drop probability can be easily changed, we denote it as a condition for network function. Based on the value of $p_{\text{drop}}$, certain network features are dropped with probability $p_{\text{drop}}$. Note that we set the default drop probability to zero for convenience. Then, loss for drop-based regularization loss with probability $p \in [0, 1]$ is

$$-\frac{1}{N} \sum_i^N \mathbf{y}_i \log\left(\sigma(f_\theta(\mathbf{x}_i|p_{\text{drop}} = p))\right). \tag{2}$$

Typically, a network with drop-based regularization is trained with equation 2. But, we conjecture that training using equation 2 with high probability $p$ may interfere with loss convergence and induce instability in training. To ensure training stability, we utilize the model output of equation equation 1, $f_\theta(\mathbf{x}_i|p_{\text{drop}} = 0)$, as guidance for drop-based regularization $f_\theta(\mathbf{x}_i|p_{\text{drop}} = p)$ instead of $\mathbf{y}_i$. In other words, equation 2 is changed as

$$-\frac{1}{N} \sum_i^N \sigma(f_\theta(\mathbf{x}_i|p_{\text{drop}} = 0)) \log\left(\sigma(f_\theta(\mathbf{x}_i|p_{\text{drop}} = p))\right). \tag{3}$$

In our Augmenting Sub-model (AugSub), the average of equation 1 and equation 3 is used as a loss function for the network. We designate $f_\theta(\mathbf{x}_i|p_{\text{drop}} = 0)$ as the main model and $f_\theta(\mathbf{x}_i|p_{\text{drop}} = p)$ as the sub-model. This naming convention is employed because a network with dropped features appears to be a subset of the entire network. In equation 3, the main model output $f_\theta(\mathbf{x}_i|p_{\text{drop}} = 0)$ is used with stop-gradient. Thus, the sub-model is trained to mimic the main model, but the gradient

for the main model is independent of the sub-model. This can be interpreted as self-distillation, where knowledge is transferred from the main model to the sub-model. Also, AugSub can easily be expanded to binary cross-entropy loss by replacing the softmax function with the sigmoid function.

Algorithm 1 describes PyTorch-style pseudo-code of training with AugSub. The drop probability is put into the network input. The gradients are calculated on the average losses from the main and the sub-model. Note that AugSub does not use additional data augmentation, optimizer steps, and network parameters for the sub-model. We will demonstrate the significant performance benefits of this simple training technique.

Since the sub-model mimics the main model, it automatically controls the difficulty. If the main model produces output closely aligned with the ground-truth label, the sub-model loss aims to attain an accurate classification output under drop-based regularization. Conversely, if the main model fails to converge, the sub-model loss becomes considerably easier than constructing a ground-truth label. In summary, AugSub prioritizes the learning process, ensuring that drop-based regularization is exclusively applied to images that produce successful output in a standard setting. We assert that the prioritized loss mechanism of AugSub enables the network to preserve its convergence speed and learning stability while maintaining the benefits of drop-based regularization.

---

**Algorithm 1** AugSub in PyTorch-style pseudo-code

```
# For drop probability p
for (x, y) in loader:
    o1, o2 = f(x, 0), f(x, p)
    loss = CE(o1, y)
    loss += CE(o2, softmax(o1.detach()))
    (loss/2).backward()
    optimizer.step()
```

---

### 3.2 DROP-BASED SUB-MODEL REGULARIZATIONS

We select three drop-based techniques for AugSub: dropout (Srivastava et al., 2014), drop-path (Huang et al., 2016), and random masking (He et al., 2022). All methods are to drop a certain intermediate feature of the network. Drop-based techniques can easily adjust the strength by controlling drop probability and are also widely used as an essential regularization in training recipes (Wightman et al., 2021; Touvron et al., 2021a; 2022a; Liu et al., 2021). In this section, we explain each drop-based techniques and our implementation.

**AugDrop.** Dropout (Srivastava et al., 2014) is a fundamental activation drop method. Dropout drops feature elements with a fixed probability. Since dropout is not related to feature structure, every element of the feature has independent drop probability $p$. Although dropout is not preferred in recent training recipes (Touvron et al., 2021a; 2022a), it is effective in the sub-model framework. For AugDrop, dropout is used for every self-attention and MLP block following the famous implementation (Wightman, 2019).

**AugPath.** Drop-path (Huang et al., 2016; Fan et al., 2019) randomly drops a total feature of the network block with a probability $p$. When a layer is dropped, the signal proceeds to the next layer through the residual path only, acting as if the dropped layer does not exist for a specific image input. Drop-path is used to adjust the regularization strength (Touvron et al., 2022a). For AugPath, we maintain drop-path of the training recipe in the main model and increase drop-path probability with a fixed rate, +0.1 or +0.2, in the sub-model.

**AugMask.** Random masking is an augmentation technique for BERT-like self-supervised learning (Bao et al., 2021; He et al., 2022). It drops input patches with a fixed ratio and uses the remaining patches as the network inputs. Despite the successes of random masking in self-supervised learning, it is often deemed too rigorous for supervised learning. Thus, we employ it in AugSub, which is designed to mitigate the intensity of regularization. We implement AugMask using MAE style token drop (He et al., 2022), allowing us to inherit the computational cost reduction by skipping network computation for the masked region.

### 3.3 ANALYSIS

We analyze the impact of AugSub with training ViT-B for 100 epochs on ImageNet-1k (Deng et al., 2009). Three variants of AugSub are applied: AugDrop, AugPath, and AugMask. Based on DeiT-III (Touvron et al., 2022a), we shorten the epoch to 100 epochs and use image resolution $224 \times 224$.

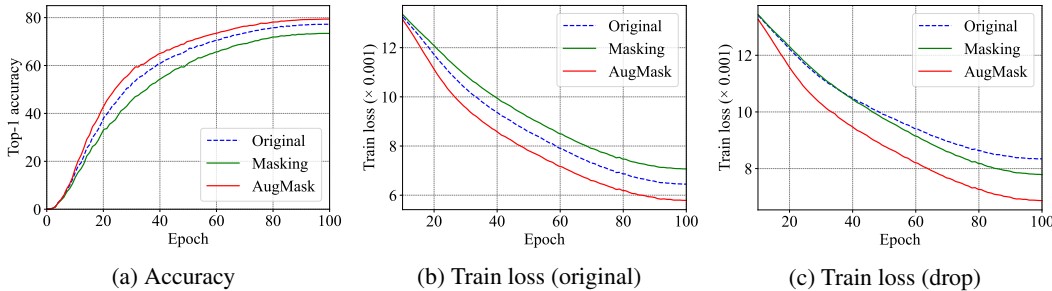

(a) Accuracy   (b) Train loss (original)   (c) Train loss (drop)

Figure 2: **Training metric analysis**. We use 50%-random masking to compare three training settings: original (equation 1), masking (equation 2), and AugMask. We visualize (a) accuracy on the validation set; (b) train loss without drop (masking); (c) train loss with drop (masking).

Table 1: **Analysis on drop regularization with/without AugSub.** The table shows 100 epochs of the ViT-B performance trained with drop regularization. Note that training loss scale $10^{-3}$ is omitted for simplicity. The table presents the average values over three separate runs, and the standard deviations are reported in Table A.4

|  | Drop ratio | Single model | | | Augmenting Sub-model (AugSub) | | |
|---|---|---|---|---|---|---|---|
|  |  | Accuracy | Train loss (original) | Train loss (drop) | Accuracy | Train loss (original) | Train loss (drop) |
| Original | - | 77.4 | 6.42 | - | - | - | - |
| Dropout | 0.1 | 76.1 | 6.60 | 6.87 | 79.1 | 5.88 | 6.32 |
|  | 0.2 | 74.1 | 6.95 | 7.34 | 79.1 | 5.82 | 6.57 |
|  | 0.3 | 71.6 | 8.34 | 7.79 | 79.1 | 5.84 | 6.90 |
| Drop-path | 0.1 | 77.4 | 6.42 | 6.42 | 78.4 | 6.11 | 6.11 |
|  | 0.2 | 74.9 | 6.74 | 7.19 | 78.7 | 5.91 | 6.48 |
|  | 0.3 | 71.6 | 7.31 | 8.04 | 78.8 | 5.87 | 7.02 |
| Masking | 25% | 76.3 | 6.60 | 6.96 | 79.0 | 5.89 | 6.38 |
|  | 50% | 73.8 | 7.02 | 7.77 | **79.4** | 5.81 | 6.89 |
|  | 75% | 67.3 | 8.08 | 9.27 | 79.2 | 5.84 | 8.15 |

We compare three settings: original, drop-based technique, and AugSub. The original uses equation 1 as the training loss, and none of the drop-based techniques is not used. For the drop-based techniques, the network is trained with equation 2. Note that it is a common practice to use drop-based techniques. We compare those two settings with AugSub. For analysis, we measured equation 1 'train loss - original' and equation 2 'train loss - drop' regardless of loss used for training. It shows how losses changed by training setting.

Figure 2 shows loss and accuracy trends in masking 50% case. When random masking is applied to training (green), loss with masking (Figure 2c) converges better than the original (blue). However, it significantly degrades the original train loss (Figure 2b), resulting in degrade in accuracy (Figure 2a). Regularization over the balance point often causes malicious effects on original train loss, which decreases accuracy. As shown in Figure 2b and 2c, AugMask improves the loss convergence for both losses, which makes a significant improvement in accuracy.

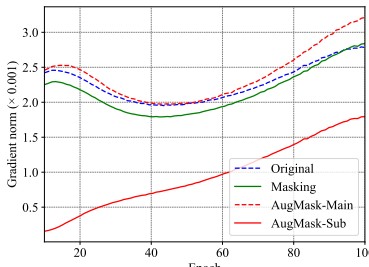

Figure 3: **Gradients magnitude.** The gradient norm is averaged value over all parameters for each epoch.

Figure 3 explains the function of AugSub loss (equation 3) in the aspect of gradients magnitude for masking 50% case. The gradient magnitude from the main model (AugMask-Main) is similar to that of other training. In contrast, gradients from the sub-model (AugMask-Sub) have a small magnitude at the early stage. As the learning progresses, the gradients from the sub-model increase. It shows that AugSub trains the network following our intention. During the early stage of training, the gradients from the main model lead the training. Following the progress of the main model training, the sub-model adaptively increases its gradient magnitude and produces a reasonable amount of gradients at the end of training.

Table 2: **Comparison of three variants of AugSub.** We use 400 epochs training with DeiT-III (Touvron et al., 2022a) to compare the performance of three drop-based regularizations integrated with AugSub: AugDrop, AugPath, and AugMask. The table presents the average over three runs. The std values are shown in Table A.5

| Architecture | Baseline | AugDrop | AugPath | AugMask |
|---|---|---|---|---|
| ViT-S/16 | 80.4 | 80.6 (+0.2) | 80.8 (+0.4) | **81.1** (+0.7) |
| ViT-B/16 | 83.5 | 83.8 (+0.3) | 83.8 (+0.3) | **84.1** (+0.6) |
| Computational costs | ×1.0 | ×2.0 | ×2.0 | ×1.5 |

Table 3: **Training from scratch with ViT using DeiT-III.** AugMask (50%) is applied to the ViT training recipe (Touvron et al., 2022a) on ImageNet-1k. Note that the training settings are identical to the original ones.

| Architecture | # params | FLOPs | 400 epochs | | 800 epochs | |
|---|---|---|---|---|---|---|
| | | | DeiT-III | + AugMask | DeiT-III | + AugMask |
| ViT-S/16 | 22.0 M | 4.6 G | 80.4 | **81.1** (+0.7) | 81.4 | **81.7** (+0.3) |
| ViT-B/16 | 86.6 M | 17.5 G | 83.5 | **84.1** (+0.6) | 83.8 | **84.2** (+0.4) |
| ViT-L/16 | 304.4 M | 61.6 G | 84.5 | **85.2** (+0.7) | 84.9 | **85.3** (+0.4) |
| ViT-H/14 | 632.1 M | 167.4 G | 85.1 | **85.7** (+0.6) | 85.2 | **85.7** (+0.5) |

Table 1 shows results of other drop-based regularization with two drop-ratio. 'Single model' represents a training with drop loss function equation 2, when additional drop-based technique is directly applied to the main model. 'Augmenting Sub-model (AugSub)' shows the performance when drop-based regularization is applied through AugSub. Similar to Figure 2, 'Train loss (original)' shows equation 1 and 'Train loss (drop)' represents loss with drop as equation 2. The results demonstrate that AugSub improves training in all three drop-based regularization cases. In all cases, AugSub improves original and drop loss convergence, which is connected to superior accuracy compared to original training.

## 4 EXPERIMENTS

We validate the effectiveness of Augmenting Sub-model (AugSub) by applying it to diverse training scenarios. We claim AugSub is an easy plug-in solution for various training recipes. Thus, we strictly follow the original training recipe, including optimizer parameters, learning rate and weight-decay, and regularization parameters. The only difference between baseline and AugSub is the drop-based technique for the sub-model. We consider AugMask our representative method among the three variants of AugSub with its cost efficiency and impressive performance. We mainly report the results with AugMask with a fixed masking ratio 50% across all experiments. Note that comparisons in the same training computation costs are reported in Table A.1 in Sec. A.1.

### 4.1 TRAINING FROM SCRATCH

The training recipe in ViTs is a key factor enabling ViT to surpass CNN; thus, the ViT training recipe is a significant and active research topic. We use a state-of-the-art ViT recipe, DeiT-III (Touvron et al., 2022a), as a baseline. Integrating additional techniques into the DeiT-III is a significant challenge, and improvements made over DeiT-III can be considered a novel state-of-the-art in ViT training.

We measure the performance of all three variants of AugSub on a 400 epochs training with Deit-III. We use Dropout (0.2), Drop-path (base + 0.1), and Masking (50%) for AugDrop, AugPath, and AugMask, respectively. Table 2 shows the results. All three variants of AugSub outperform the baseline. Among the three methods, AugMask shows the best performance. Also, AugMask has the lowest computation costs due to MAE (He et al., 2022)-style computation reduction. Thus, we conclude that AugMask (50%) is the best in practice for other training recipes.

We expand the experiment with AugMask (50%). Various sizes of ViTs are trained with AugMask (50%) on 400 and 800 epochs training. The results are shown in Table 3. AugMask significantly improves performance in all settings. In 400 epochs training, AugMask improves DeiT-III with substantial margins, which even outperforms 800 epochs training except for ViT-S/16. AugMask also demonstrates superior performance in 800 epochs of training. The impact of AugMask is impressively sustained even for larger models like ViT-L/16 and ViT-H/16. It is worth noting that

Table 4: **ImageNet-1k finetuning.** We report finetuned performance of MAE (He et al., 2022), BEiT v2 (Peng et al., 2022) and CLIP finetuning (Dong et al., 2022) with AugMask (50%). Official pretrained weights are used.

| Pretraining | Finetuning recipe | Finetuning epochs | Architecture | Baseline | +AugMask |
|---|---|---|---|---|---|
| MAE (1600 epochs) | MAE finetuning (+ AugMask) | 100 | ViT-B/16 | 83.6 | **83.9** (+0.3) |
| | | 50 | ViT-L/16 | 85.9 | **86.1** (+0.2) |
| | | 50 | ViT-H/14 | 86.9 | **87.2** (+0.3) |
| BEiT v2 (1600 epochs) | BEiT v2 finetuning (+ AugMask) | 100 | ViT-B/16 | 85.5 | **85.6** (+0.1) |
| | | 50 | ViT-L/16 | 87.3 | **87.4** (+0.1) |
| CLIP | Finetuning CLIP (+ AugMask) | 50 | ViT-B/16 | 84.8 | **85.2** (+0.4) |
| | | 30 | ViT-L/14 | 87.5 | **87.8** (+0.3) |

Table 5: **ImageNet-1k with hierarchical architecture.** We show the performance of ResNet (He et al., 2016) and Swin Transformer (Liu et al., 2021) trained from scratch with AugMask (50%).

| Training recipe | Epochs | Architecture | # params | FLOPs | Baseline | + AugMask |
|---|---|---|---|---|---|---|
| ResNet-RSB A2 | 300 | ResNet50 | 25.6 M | 4.1 G | 79.7 | **80.0** (+0.3) |
| | | ResNet101 | 44.5 M | 7.9 G | 81.4 | **82.1** (+0.7) |
| | | ResNet152 | 60.2 M | 11.6 G | 81.8 | **82.8** (+1.0) |
| Swin Transformer | 300 | Swin-T | 28.3 M | 4.5 G | 81.3 | **81.4** (+0.1) |
| | | Swin-S | 49.6 M | 8.7 G | 83.0 | **83.4** (+0.4) |
| | | Swin-B | 87.9 M | 15.4 G | 83.5 | **83.9** (+0.4) |

ViT-H + AugMask (400 epochs) outperforms ViT-H/16 (800 epochs) with a significant +0.5pp gain even with half training length. Thus, AugMask is an effective way to improve ViT training.

## 4.2 FINETUNING

Following the emergence of self-supervised learning on ImageNet (Deng et al., 2009), the significance of finetuning has notably increased. Generally, self-supervised learning, such as MAE (He et al., 2022) and BEiT (Bao et al., 2021; Peng et al., 2022), does not use supervised labels at pretraining, which makes AugSub inapplicable for pretraining. However, most methods utilize supervised finetuning steps after pretraining to demonstrate their performance. Thus, we apply our AugMask (50%) to the finetuning stage. Note that we strictly follow original finetuning recipes and apply AugMask (50%) based on it. All finetuning is conducted using officially released pretrained weights.

We utilize three finetuning recipes: MAE (He et al., 2022), BEiT v2 (Peng et al., 2022), and Finetune CLIP (Dong et al., 2022). MAE (He et al., 2022) is a representative method of masked image models (MIM). Since our random masking is motivated by MAE, AugMask is seamlessly integrated into MAE finetuning process. BEiT v2 (Peng et al., 2022) uses pretrained CLIP for MIM and achieves superior performance compared to MAE. Following the masking strategy of BEiT v2 using mask-token, we adjust AugMask to masking using mask-token from the pretrained weight. Finetune CLIP (Dong et al., 2022) is a finetuning recipe for CLIP (Radford et al., 2021) pretrained weights. AugMask is applied to finetuning CLIP without change, the same as MAE finetune.

Table 4 shows the finetuning results. AugMask improves the performance of all finetune recipes, including large-scale ViT models. This is notable as it shows substantial improvement with a short finetuning phase of fewer than 100 epochs compared to the pretraining period of 1600 epochs. In MAE finetuning, AugMask improves 0.2 - 0.3pp in all model sizes. AugMask is also effective on BEiT v2, which utilizes RPE (Liu et al., 2021; Bao et al., 2021) and masking strategy with mask-token. Even in CLIP finetuning, AugMask achieves substantial improvement. In finetuning CLIP, we report performance at the last epoch rather than selecting the best performance in early epochs. The best performance of finetuning CLIP with AugMask is the same as the baseline.

## 4.3 HIERARCHICAL ARCHITECTURE

We extend experiments to architectures with hierarchical spatial dimensions: ResNet (He et al., 2016) and Swin Transformer (Liu et al., 2021). Unlike ViT maintains spatial token length for all

Table 6: **Comparison in ImageNet-1k training.** The table shows performance and GPU costs of augmenting methods for ViT-B 400 epoch training.

| Method | Accuracy | GPU days |
|---|---|---|
| Original (2022a) | 83.5 | 17.3 |
| RepeatedAug (2020) | 83.5 (+0.0) | 36.8 (+113%) |
| GradAug (2020) | 83.2 (-0.3) | 39.7 (+129%) |
| CoSub (2022b) | 83.9 (+0.4) | 35.3 (+104%) |
| AugMask | **84.1** (+0.6) | **25.1** (+45%) |

Table 7: **Comparison in MAE finetuning.** The table displays performances and GPU costs of augmenting methods when applied to MAE ViT-B fine-tuning.

| Method | Accuracy | GPU days |
|---|---|---|
| Original (2022a) | 83.6 | 6.0 |
| RepeatedAug (2020) | 83.1 (-0.5) | 11.8 (+97%) |
| GradAug (2020) | **84.0** (+0.4) | 23.4 (+290%) |
| CoSub (2022b) | 83.8 (+0.2) | 11.0 (+83%) |
| AugMask | 83.9 (+0.3) | **8.6** (+43%) |

layers, those networks change the spatial size of features in the middle of layers, requiring a change in masking strategy. We apply AugMask (50%) to ResNet and Swin Transformer. We simply fill out masked regions with zero pixels for ResNets and replace masked regions to mask-tokens for Swin Transformer. It maintains the spatial structure and enables spatial size reduction of hierarchical architecture. Following previous literature (Woo et al., 2023), we use random masking with patch-size $32 \times 32$. Note that the computation reduction of AugMask is not applicable for this case due to changes in the masking strategy. Thus, AugMask costs double the training budget. For ResNet, we use a high-performance training recipe (Wightman et al., 2021) with 300 epochs. The recipe of original paper (Liu et al., 2021) is used for the Swin Transformer training. We strictly follow the training recipe and apply AugMask without recipe tuning.

Results are shown in Table 5. AugMask achieves impressive performance gains even in ResNet and Swin Transformer. ResNet is a convolutional neural network using Batch Normalization (Ioffe & Szegedy, 2015). Thus, it is substantially different from ViT. Swin Transformer uses a different training recipe from the conventional ViT recipe (Touvron et al., 2022a). Thus, an improvement on Swin shows that AugMask can be used for different training from scratch recipes without tuning. In summary, the effectiveness of AugMask is not limited to ViT architectures and is applicable to hierarchical architectures.

### 4.4 COMPARISON WITH OTHER AUGMENTING METHODS

We compare AugMask with other augmenting methods: RepeatedAug (Hoffer et al., 2020), GradAug (Yang et al., 2020), and CoSub (Touvron et al., 2022b). RepeatedAug (2020) is a data augmentation technique widely used for ViT training. For comparison, we doubled the RepeatedAug and compared the performance. GradAug (2020) is an early-stage sub-network augmentation that utilizes network slimming (Yu et al., 2018) to build sub-network. CoSub introduces a sub-network based on drop-path (Fan et al., 2019) and uses the sub-network as co-training (Zhang et al., 2018). We compared AugMask (AugSub with random masking) with those methods as a general augmentation technique. Comparisons include two training scenarios: 400 epochs ImageNet-1k training from scratch (Touvron et al., 2022a) and MAE fine-tuning on ImageNet-1k (He et al., 2022). All experiments are conducted on a V100×8 machine. We compare top-1 accuracy and training computation cost of each augmentation method. The computation is reported as the number of days when it is trained with a single V100 machine.

Table 6 shows the 400 epoch training from scratch result, and Table 7 represents MAE fine-tuning result. Note that GradAug in Table 6 is 200 epochs training result to adjust computation cost similar to other methods. All augmentation methods require additional computation costs. In particular, GradAug spends almost 300% of additional training costs compared to original methods. On the other hand, our AugMask only requires a small amount of additional costs (below 50%), which is a significant advantage compared to other methods. With the smallest computation, our AugMask achieves substantial performance improvements. AugMask performs superior to CoSub in all cases and performs similarly to GradAug in MAE fine-tuning with 1/6 training costs.

### 4.5 ROBUSTNESS

We evaluate the impact of AugMask in various robustness benchmarks. We use models trained for 800 epochs in Table 3. Table 8 shows the results. ViT models trained with AugMask demonstrate superior performance in all robustness metrics. AugMask outperforms the baseline in natural adversarial

Table 8: **Robustness benchmark.** Table shows the robustness benchmark for ViT pretrained with/without AugMask. In all metrics, higher scores indicate better results.

| Model | +AugMask | IN-1k | IN-V2 | IN-Real | IN-A | IN-R | ObjNet | SI-size | SI-loc | SI-rot |
|-------|----------|-------|-------|---------|------|------|--------|---------|--------|--------|
| ViT-S | - | 81.4 | 70.1 | 87.0 | 23.4 | 46.4 | 32.6 | 55.0 | 39.8 | 37.8 |
|       | ✔ | **81.7** | **71.0** | **87.4** | **26.9** | **47.2** | **33.5** | **56.7** | **42.5** | **39.9** |
| ViT-B | - | 83.8 | 73.4 | 88.2 | 36.8 | 54.1 | 35.7 | 58.0 | 42.7 | 41.5 |
|       | ✔ | **84.2** | **74.0** | **88.6** | **41.9** | **54.4** | **37.2** | **59.0** | **44.8** | **43.3** |
| ViT-L | - | 84.9 | 74.8 | 88.8 | 45.3 | 57.4 | 38.8 | 59.8 | 46.5 | 45.0 |
|       | ✔ | **85.3** | **75.8** | **89.2** | **51.1** | **58.5** | **40.0** | **60.2** | **46.8** | **45.9** |
| ViT-H | - | 85.2 | 75.7 | 89.2 | 51.9 | 58.8 | 40.1 | 61.9 | 49.0 | 46.8 |
|       | ✔ | **85.7** | **76.5** | **89.6** | **58.3** | **59.9** | **41.7** | **62.4** | **50.1** | **48.4** |

Table 9: **Transfer learning to small-scale datasets.** Table shows transfer learning performance with/without AugMask. We measure the performance when AugMask is applied to pretraining and finetuning. The table presents the average values over three separate runs, and the standard deviations are reported in Table A.6

| Model | Pretraining + AugMask | Finetuning + AugMask | CIFAR-10 | CIFAR-100 | Flowers | Cars | iNat-18 | iNat-19 |
|-------|------------------------|----------------------|----------|-----------|---------|------|---------|---------|
| ViT-S/16 | - | - | 98.8 | 90.0 | 94.5 | 80.9 | 70.1 | 76.7 |
|          | ✔ | - | **98.9** | **90.6** | 95.2 | 81.2 | 70.8 | 77.0 |
|          | ✔ | ✔ | 98.8 | 89.9 | **98.3** | **92.2** | **71.2** | **77.1** |
| ViT-B/16 | - | - | 99.1 | 91.7 | 97.5 | 90.0 | 73.2 | 78.5 |
|          | ✔ | - | **99.2** | **91.9** | 97.7 | 90.2 | 73.6 | 78.8 |
|          | ✔ | ✔ | 98.8 | 89.6 | **98.7** | **92.8** | **73.9** | **79.1** |

examples (Hendrycks et al., 2021b) (IN-A), objects in different styles and textures (IN-R (Hendrycks et al., 2021a)), controls in rotation, background, and viewpoints (ObjNet (Barbu et al., 2019)), and SI-scores (Djolonga et al., 2021) (SI-size, SI-loc, and SI-rot). The results demonstrate that the improvement of AugMask is not limited to ImageNet validation and has been verified across various robustness metrics.

## 4.6 TRANSFER LEARNING

Improvement on pretraining can boost the performance of downstream tasks (Kornblith et al., 2019). We measure transfer learning performance of AugMask using 800 epochs pretrained weight from Table 3. We use the CIFAR-10 (Krizhevsky et al., 2009), CIFAR-100 (Krizhevsky et al., 2009), Oxford Flowers-102 (Nilsback & Zisserman, 2008), Stanford Cars (Krause et al., 2013) and iNaturalist (Van Horn et al., 2018) datasets. We use AdamW training recipe (Touvron et al., 2022a). We also evaluate performance when AugMask (50%) is applied to the finetuning process. Table 9 shows the results. The backbone pretrained with AugMask consistently outperforms the DeiT-III backbone across all cases. Moreover, when AugMask is applied to the finetuning process, it further boosts performance in most cases except CIFAR.

## 5 CONCLUSION

In this work, we have presented a new method for additional regularization across various training recipes. Our method, Augmented Sub-model (AugSub), is designed to leverage drop-based regularization within a sub-model, which is separated from main training and uses a relaxed loss function. Our extensive analysis reveals that AugSub effectively mitigates malicious effects of additional regularization while accelerating the convergence speed, yielding superior performance. We verify AugSub on various training recipes, including diverse architecture. Notably, AugMask, a specific implementation of AugSub for random masking, demonstrates significant performance improvements across diverse scenarios. We claim that AugSub is a substantial advancement in training recipes and contributes to building novel regularization strategies.

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

APPENDIX

# A  EXPERIMENTS

## A.1  TRAINING BUDGET

We have shown that AugMask effectively improves the performance of various architectures. However, AugMask requires additional computation costs for the sub-model, which increases training costs. Thus, we analyze AugMask regarding its training costs to determine if AugMask could be an effective solution within a limited training budget. We compare AugMask with training recipes set to $\times 1.5$ epochs to align with the training budget. The training budget is quantified regarding required GPU days when only a single NVIDIA V100 GPU is used for training. Table A.1 shows the results. In DeiT-III (Touvron et al., 2022a) training, AugMask outperforms baseline with $\times 1.5$ epochs setting. Thus, AugMask is superior to the long epoch training to spend computation costs for training ViT. MAE finetuning with $\times 1.5$ epochs even degrades the performance compared to the baseline. For ResNet, we compare 300 epochs AugMask with 600 epochs training recipe RSB (Wightman et al., 2021) A1. AugMask outperforms 600 epochs training recipes in ResNet101 and ResNet152. Consequently, the results show that AugMask is an effective way to improve training, even considering computation costs for the sub-model.

Table A.1: **Comparison in the same training budget.** All training has been conducted with NVIDIA V100 8 GPUs. GPU days refer to the number of days required for training when using a single V100 GPU.

|  | Architecture | Training recipe | +AugMask | Epochs | GPU days | Accuracy |
|---|---|---|---|---|---|---|
| DeiT-III Training | ViT-S/16 | DeiT-III | - | 600 | 22 | 80.7 |
|  |  |  | ✔ | 400 | 22 | **81.2** (+0.5) |
|  |  | DeiT-III | - | 1200 | 45 | 81.6 |
|  |  |  | ✔ | 800 | 44 | **81.7** (+0.1) |
|  | ViT-B/16 | DeiT-III | - | 600 | 26 | 83.7 |
|  |  |  | ✔ | 400 | 25 | **84.1** (+0.4) |
|  |  | DeiT-III | - | 1200 | 52 | 83.8 |
|  |  |  | ✔ | 800 | 50 | **84.2** (+0.4) |
| MAE Finetuning | ViT-B/16 | MAE Finetune | - | 150 | 9 | 83.5 |
|  |  |  | ✔ | 100 | 9 | **83.9** (+0.4) |
|  | ViT-L/16 | MAE Finetune | - | 75 | 14 | 85.5 |
|  |  |  | ✔ | 50 | 14 | **86.1** (+0.6) |
| ResNet Training | ResNet50 | RSB A1 | - | 600 | 22 | 80.4 |
|  |  | RSB A2 | ✔ | 300 | 14 | **80.0** (-0.4) |
|  | ResNet101 | RSB A1 | - | 600 | 24 | 81.5 |
|  |  | RSB A2 | ✔ | 300 | 20 | **82.1** (+0.6) |
|  | ResNet152 | RSB A1 | - | 600 | 32 | 82.0 |
|  |  | RSB A2 | ✔ | 300 | 29 | **82.8** (+0.8) |

## A.2  DOWNSTREAM TASKS

**Semantic segmentation.** Using the segmentation recipe of BEiT v2 (Peng et al., 2022), we train UpperNet (Xiao et al., 2018) with ViT backbone on ADE-20k (Zhou et al., 2017) dataset. Table A.2 shows the results in two ways: single-scale and multi-scale evaluation. On both evaluations, the backbone pretrained with AugMask demonstrates superior performance in ViT-B and ViT-L.

**Object detection and instance segmentation.** We utilize Cascaded Mask R-CNN (Cai & Vasconcelos, 2019) with ViT backbones (Li et al., 2022) for MS COCO (Lin et al., 2014), which conducts object detection and instance segmentation simultaneously. ViTDet (Li et al., 2022) is used as a training recipe for this experiment. Table A.3 shows the results. The metric $AP^{box}$ quantifies the performance in object detection, while $AP^{mask}$ provides performance in instance segmentation. In both measures, the backbone pretrained with AugMask outperforms the DeiT-III backbone.

Table A.2: **Semantic segmentation on ADE-20k.** UpperNet for ViT backbone is trained with the BEiTv2 segmentation recipe.

|  | Single-scale mIoU | | Multi-scale mIoU | |
|---|---|---|---|---|
|  | DeiT-III | + AugMask | DeiT-III | + AugMask |
| ViT-B | 48.8 | **49.4** (+0.6) | 49.7 | **50.2** (+0.5) |
| ViT-L | 51.7 | **52.2** (+0.5) | 52.3 | **52.7** (+0.4) |

Table A.3: **Detection and instance segmentation on MS COCO.** Cascaded Mask R-CNN with ViT-B is used.

|  | $AP^{box}$ | $AP^{mask}$ |
|---|---|---|
| DeiT-III | 50.7 | 43.6 |
| +AugMask | **50.9** (+0.2) | **43.9** (+0.3) |

### A.3 MEAN AND STANDARD DEVIATION

We provide mean and standard deviation for experiments using different random seeds. The values presented in this section are the result of three independent runs with different seeds.

Table A.4 shows the mean and standard deviation values for Table 1 in the original paper. Note that some numbers changed from the original table due to a minor bug fixing in the analysis code. Table A.4 shows the superiority of our AugSub in additional regularization training.

Table A.5 shows 400 epochs training with DeiT-III (Touvron et al., 2022a), which is reported in Table 2 of the paper. Our AugSub improves the performance of ViT training with three regularizations: AugDrop, AugPath, and AugMask. Among the three variants, AugMask demonstrates the best performance.

We measure mean and standard deviation values for transfer learning experiments (Table 9 in the paper) as shown in Table A.6. The random masking variant of our method (AugMask) demonstrates significant gains, which surpass the standard deviation of performance. Table A.7 presents short training (300 epochs) results. AugMask shows substantial improvements when it is applied to both pretraining and finetuning processes.

Table A.4: **Mean and std for analysis on drop regularization with/without AugSub.** The table shows 'mean $\pm$ std' values for experiments in Table 1 of the paper. Note that training loss scale $10^{-3}$ is omitted for simplicity.

|  | Drop ratio | Single model | | | Augmenting Sub-model (AugSub) | | |
|---|---|---|---|---|---|---|---|
|  |  | Accuracy | Train loss (original) | Train loss (drop) | Accuracy | Train loss (original) | Train loss (drop) |
| Original | - | 77.40 ± 0.20 | 6.42 ± 0.03 | - | - | - | - |
| Dropout | 0.1 | 76.09 ± 0.25 | 6.60 ± 0.07 | 6.87 ± 0.06 | 79.14 ± 0.15 | 5.88 ± 0.02 | 6.32 ± 0.02 |
|  | 0.2 | 74.10 ± 0.22 | 6.95 ± 0.06 | 7.34 ± 0.06 | 79.10 ± 0.11 | 5.82 ± 0.04 | 6.57 ± 0.04 |
|  | 0.3 | 71.62 ± 0.29 | 8.34 ± 0.03 | 7.79 ± 0.03 | 79.09 ± 0.15 | 5.84 ± 0.03 | 6.90 ± 0.03 |
| Drop-path | 0.1 | 77.40 ± 0.20 | 6.42 ± 0.03 | 6.42 ± 0.03 | 78.36 ± 0.03 | 6.11 ± 0.01 | 6.11 ± 0.01 |
|  | 0.2 | 74.92 ± 0.12 | 6.74 ± 0.04 | 7.19 ± 0.03 | 78.72 ± 0.12 | 5.91 ± 0.01 | 6.48 ± 0.01 |
|  | 0.3 | 71.57 ± 0.10 | 7.31 ± 0.02 | 8.04 ± 0.02 | 78.80 ± 0.15 | 5.87 ± 0.02 | 7.02 ± 0.01 |
| Masking | 25% | 76.33 ± 0.28 | 6.60 ± 0.05 | 6.96 ± 0.05 | 79.02 ± 0.12 | 5.89 ± 0.03 | 6.38 ± 0.04 |
|  | 50% | 73.78 ± 0.08 | 7.02 ± 0.04 | 7.77 ± 0.03 | 79.36 ± 0.01 | 5.81 ± 0.01 | 6.89 ± 0.01 |
|  | 75% | 67.27 ± 0.25 | 8.08 ± 0.05 | 9.27 ± 0.04 | 79.16 ± 0.05 | 5.84 ± 0.01 | 8.15 ± 0.02 |

Table A.5: **Mean and std for three variants of AugSub.** We report 'mean $\pm$ std' values for 400 epochs training with DeiT-III (Touvron et al., 2022a). Note that we use the performance of original paper (Touvron et al., 2022a) for baseline training.

| Architecture | Baseline | AugDrop | AugPath | AugMask |
|---|---|---|---|---|
| ViT-S/16 | 80.40 ± 0.33 | 80.57 ± 0.12 | 80.78 ± 0.05 | 81.08 ± 0.12 |
| ViT-B/16 | 83.46 ± 0.04 | 83.83 ± 0.11 | 83.80 ± 0.12 | 84.08 ± 0.02 |

## B IMPLEMENTATION DETAILS

Most experiments in the paper were performed on a machine with NVIDIA V100 8 GPUs. The exceptions were DeiT-III (Touvron et al., 2022a) experiments for ViT-L and ViT-H in Table 3 and

Table A.6: **Mean and std for transfer learning to small scale datasets.** Table shows 'mean $\pm$ std' values for transfer learning performance with/without AugMask. We measure the performance when AugMask is applied to pretraining and finetuning.

| Model | Pretraining + AugMask | Finetuning + AugMask | CIFAR10 | CIFAR100 | Flowers | Cars | iNat-18 | iNat-19 |
|---|---|---|---|---|---|---|---|---|
| ViT-S | - | - | $98.83 \pm 0.05$ | $89.96 \pm 0.15$ | $94.54 \pm 1.71$ | $80.86 \pm 0.71$ | $70.12 \pm 0.13$ | $76.69 \pm 0.56$ |
| | ✔ | - | $98.88 \pm 0.09$ | $90.63 \pm 0.09$ | $95.19 \pm 1.95$ | $81.23 \pm 0.73$ | $70.82 \pm 0.03$ | $77.00 \pm 0.21$ |
| | ✔ | ✔ | $98.77 \pm 0.05$ | $89.87 \pm 0.17$ | $98.25 \pm 0.51$ | $92.17 \pm 0.14$ | $71.17 \pm 0.21$ | $77.12 \pm 0.48$ |
| ViT-B | - | - | $99.07 \pm 0.05$ | $91.69 \pm 0.15$ | $97.52 \pm 0.51$ | $90.05 \pm 0.24$ | $73.16 \pm 0.05$ | $78.49 \pm 0.62$ |
| | ✔ | - | $99.19 \pm 0.03$ | $91.89 \pm 0.04$ | $97.73 \pm 0.30$ | $90.18 \pm 0.12$ | $73.61 \pm 0.08$ | $78.77 \pm 0.05$ |
| | ✔ | ✔ | $98.82 \pm 0.03$ | $89.55 \pm 0.05$ | $98.68 \pm 0.16$ | $92.77 \pm 0.09$ | $73.88 \pm 0.12$ | $79.07 \pm 0.55$ |

Table A.7: **Transfer learning at short (300 epochs) training.** Table shows 'mean $\pm$ std' values for transfer learning at 300 epochs. We measure the performance when AugMask is applied to pretraining and finetuning.

| Model | Pretraining + AugMask | Finetuning + AugMask | CIFAR10 | CIFAR100 | Flowers | Cars |
|---|---|---|---|---|---|---|
| ViT-S | - | - | $98.41 \pm 0.12$ | $87.27 \pm 0.19$ | $66.66 \pm 2.52$ | $46.04 \pm 3.72$ |
| | ✔ | - | $98.48 \pm 0.06$ | $87.54 \pm 0.33$ | $72.61 \pm 1.20$ | $45.46 \pm 1.80$ |
| | ✔ | ✔ | $\mathbf{98.96} \pm 0.06$ | $\mathbf{90.81} \pm 0.09$ | $\mathbf{96.64} \pm 0.23$ | $\mathbf{87.43} \pm 0.43$ |
| ViT-B | - | - | $98.97 \pm 0.10$ | $90.33 \pm 0.14$ | $90.92 \pm 1.60$ | $78.52 \pm 0.59$ |
| | ✔ | - | $99.06 \pm 0.02$ | $90.82 \pm 0.21$ | $92.45 \pm 0.90$ | $80.34 \pm 0.56$ |
| | ✔ | ✔ | $\mathbf{99.15} \pm 0.04$ | $\mathbf{91.52} \pm 0.16$ | $\mathbf{98.44} \pm 0.13$ | $\mathbf{92.22} \pm 0.03$ |

object detection in Table A.3, conducted with NVIDIA A100 64 GPUs. Also, we use a single NVIDIA V100 for transfer learning in Table 9.

We strictly follow original training recipes for experiments. We denote details of the training recipes to clarify our implementation details and assist in reproducing our results. Table B.1 shows training recipes used for Table 2, 3, 4, and 5 of the paper. It demonstrates that our AugMask is validated on various training recipes that cover diverse regularization and optimizer settings and achieves consistent improvement on all settings, which exhibits the general applicability of AugMask. Note that AugMask is applied to all recipes with the same masking ratio of 0.5. Thus, AugMask does not require hyper-parameter tuning specialized for each recipe.

Model-specific training recipes of DeiT-III (Touvron et al., 2022a) are reported in Table B.2. DeiT-III achieves strong performance with sophistically tuned training parameters mainly focused on input size and drop-path rate. It makes improving DeiT-III more challenging than other recipes, which is accomplished by our AugSub with a significant performance gap.

For semantic segmentation in ADE20k, we use BEiT v2 (Peng et al., 2022) segmentation recipe that utilizes MMCV (Contributors, 2018) and MMSeg (Contributors, 2020) library. Following the default setting, we replace the ViT backbone with the DeiT-III backbone, which includes layer-scale (Touvron et al., 2021b). Then, we train the segmentation task for 160k iteration using DeiT-III and DeiT-III + AugMask pretrained backbone.

We use Detectron2 (Wu et al., 2019) for object detection and instance segmentation task on MSCOCO (Lin et al., 2014) dataset. Among various recipes in the Detectron2 library, we use ViTDet (Li et al., 2022) as a recent and strong recipe for a ViT-based detector. We train ViTDet Cascaded Mask-RCNN with DeiT-III and DeiT-III + AugMask pretrained backbone and report performance after MSCOCO 100 epoch training.

For transfer learning, we use the AdamW training recipe on DeiT-III (Touvron et al., 2022a) transfer learning. We use AdamW with lr $10^{-5}$, weight-decay 0.05, batch size 768. Drop-path (Huang et al., 2016) and random erasing (Zhong et al., 2020) are not used. Data augmentation is set to be the same as DeiT-III, and we train ViT for 1000 epochs with a cosine learning rate decay. For CIFAR datasets, we resize $32 \times 32$ image to $224 \times 224$ to use ImageNet pretrained backbones. In the case of iNaturalist datasets (Van Horn et al., 2018), we use AdamW with lr $7.5 \times 10^{-5}$, weight-decay 0.05, batch size 768. Drop-path and random erasing ratios are set to 0.1, and ViT is trained for 360 epochs with cosine learning rate decay.

Table B.1: **Details of various training recipes used for experiments.** Our AugMask achieves consistent improvement in all training recipes that covers diverse regularization and optimizer settings.

| Training recipe | DeiT-III | RSB A2 | Swin | MAE | BEiT v2 | FT-CLIP |
|---|---|---|---|---|---|---|
| Fine-tuning | ✗ | ✗ | ✗ | ✔ | ✔ | ✔ |
| Epoch | 400 / 800 | 300 | 300 | 100, 50 | 100, 50 | 50, 30 |
| Batch size | 2048 | 2048 | 1024 | 1024 | 1024 | 2048 |
| Optimizer | LAMB | LAMB | AdamW | AdamW | AdamW | AdamW |
| LR | $3 \times 10^{-3}$ | $5 \times 10^{-3}$ | $1 \times 10^{-3}$ | $(2, 4) \times 10^{-3}$ | $(5, 2) \times 10^{-4}$ | $(6, 4) \times 10^{-4}$ |
| LR decay | cosine | cosine | cosine | cosine | cosine | cosine |
| Layer LR decay | - | - | - | 0.65, 0.75 | 0.6, 0.8 | 0.6, 0.65 |
| Weight decay | 0.03 / 0.05 | 0.01 | 0.05 | 0.05 | 0.05 | 0.05 |
| Warmup epochs | 5 | 5 | 20 | 5 | 20, 5 | 10, 5 |
| Loss | BCE | BCE | CE | CE | CE | CE |
| Label smoothing | - | - | 0.1 | 0.1 | 0.1 | 0.1 |
| Dropout | - | - | - | - | - | - |
| Drop-path | Table B.2 | 0.05 | 0.1 | 0.1, 0.2, 0.3 | 0.2 | - |
| Repeated aug | ✔ | ✔ | - | - | - | - |
| Gradient clip | 1.0 | - | 5.0 | - | - | - |
| RandAugment | Three Aug. | 7 / 0.5 | 9 / 0.5 | 9 / 0.5 | 9 / 0.5 | 9 / 0.5 |
| Mixup alpha | 0.8 | 0.1 | 0.8 | 0.8 | 0.8 | - |
| CutMix alpha | 1.0 | 1.0 | 1.0 | 1.0 | 1.0 | - |
| Random erasing | - | - | 0.25 | 0.25 | 0.25 | 0.25 |
| Color jitter | 0.3 | - | 0.4 | - | 0.4 | 0.4 |
| EMA | - | - | - | - | - | 0.9998 |
| Train image size | Table B.2 | $224 \times 224$ | $224 \times 224$ | $224 \times 224$ | $224 \times 224$ | $224 \times 224$ |
| Test image size | $224 \times 224$ | $224 \times 224$ | $224 \times 224$ | $224 \times 224$ | $224 \times 224$ | $224 \times 224$ |
| Test crop ratio | 1.0 | 0.95 | 0.875 | 0.875 | 0.875 | 1.0 |

Table B.2: **Model specific recipes of DeiT-III (Touvron et al., 2022a).** The table shows model-size and training length-specific training arguments used for the DeiT-III recipe. In addition to Table B.1, DeiT-III utilizes drop-path and image size to adjust the recipe for diverse model-size and training lengths.

| | | 400 epochs | | | | 800 epochs | | | |
|---|---|---|---|---|---|---|---|---|---|
| | | ViT-S | ViT-B | ViT-L | ViT-H | ViT-S | ViT-B | ViT-L | ViT-H |
| Pretraining | Image size | 224 | 192 | 192 | 160 | 224 | 192 | 192 | 160 |
| | Drop-path | 0.0 | 0.1 | 0.4 | 0.5 | 0.05 | 0.2 | 0.45 | 0.6 |
| | LR | 0.004 | 0.003 | | | 0.004 | 0.003 | | |
| | Weight decay | 0.03 | | | | 0.05 | | | |
| Resolution Finetuning | Drop-path | - | 0.2 | 0.45 | 0.55 | - | 0.2 | 0.45 | 0.55 |
| | Epochs | - | 20 | | | - | 20 | | |
| | Image size | - | 224 x 224 | | | - | 224 x 224 | | |
| | Optimizer | - | AdamW | | | - | AdamW | | |
| | LR | - | 1e-5 | | | - | 1e-5 | | |

