# OpenReview forum: "Elevating Augmentation: Boosting Performance via Sub-Model Training"
_ICLR.cc/2024/Conference — ICLR 2024 Conference Withdrawn Submission_

### Official Review · Reviewer_FBYo · 2023-10-29

**Soundness:** 2 fair
**Presentation:** 3 good
**Contribution:** 2 fair
**Rating:** 5
**Confidence:** 4

**Summary:**

This paper proposed a new regulation method to improve the performance of deep learning models. The idea is to randomly sample sub-networks from the full-network, and train the sub-networks together with the full network, with the supervision from the full network. The method achieves better performance than other regularization methods on multiple network backbones.

**Strengths:**

1.	There are comprehensive experiments and ablation study to show the effectiveness of the proposed method.
2.	The proposed method achieves better performance than other methods on different backbones, although the improvements are not that significant.
3.	The paper is well-written and easy to follow

**Weaknesses:**

1.	The idea of this paper is very similar to GradAug [1] and CoSub [2]. In GradAug, the author proposed to sample sub-networks and train them with the full-networks for improvement representation learning. The difference is that in this work the author did this based on ViT backbones. In CoSub, the author similarly proposed to sample sub-networks and train them together with the full-network. I suggest the author to conduct a thorough discussion and comparison with these methods, and give some insights on why the proposed method could be better or worse.
2.	I am not sure about the significance of the proposed method. Compared to the baseline, the proposed method mostly achieves an improvement within 0.5%, but with 1.5X training cost.
3.	Could the proposed method be applied to other tasks besides image classification? Such as detection and segmentation? If it is only applicable to image classification, it further limits the significance of the method.

**Questions:**

Please see the weakness part.

---

### Official Review · Reviewer_AdC8 · 2023-10-30

**Soundness:** 2 fair
**Presentation:** 2 fair
**Contribution:** 2 fair
**Rating:** 3
**Confidence:** 4

**Summary:**

The paper proposes a regularization method called Augmenting Aub-model (AugSub), which imposes consistency between outputs with and without a stochastic masking method (e.g., dropout, drop-path, and random masking used in mask MAE).
The paper shows using random masking, AugMask, is the most effective in terms of both efficiency and accuracy.
The proposed method constantly improves accuracy from baselines in various models and tasks.

**Strengths:**

- The method is simple and easy to use.
- According to Table 1, the method works well regardless of a drop ratio.

**Weaknesses:**

1. The method is inherently the same as the consistency regularization used in semi-supervised learning [1][2].
The difference between them is where the stochasticity is induced. As the authors discuss in the introduction, there are various regularization methods, and I wonder why the authors focus on drop-based techniques.
I think the proposed method using data augmentation or other techniques instead of drop-based methods should be compared.
In addition, I think using the ema model[3] instead of the sub-model is also an important baseline.

2. The paper shows the proposed method works well for various settings but does not explain why the proposed method works.
The proposed method can be regarded as a combination of drop-based techniques and soft targets (self-distillation).
I suspect the soft targets inherently improve accuracy because the effectiveness of the soft targets, including self-distillation and label smoothing, is verified in various tasks and models.
If so, drop-based techniques are not essential.
Thus, the author would be better to design the experiments that reveal what is essential.

[1]Oliver, Avital, et al. "Realistic evaluation of deep semi-supervised learning algorithms." 2018
[2]Sohn, Kihyuk, et al. "Fixmatch: Simplifying semi-supervised learning with consistency and confidence." 2020
[3]Tarvainen, Antti, and Harri Valpola. "Mean teachers are better role models: Weight-averaged consistency targets improve semi-supervised deep learning results." 2017

**Questions:**

- Why does the accuracy of MAE finetuning in Table A.1 decrease from Table 4? (e.g., ViT-L/16 with 50 epochs  reports 85.9 in Table 4 but ViT-L/16 with 75 epochs reports 85.5 in Table A.1.)

---

### Official Review · Reviewer_ZArF · 2023-11-01

**Soundness:** 3 good
**Presentation:** 3 good
**Contribution:** 2 fair
**Rating:** 6
**Confidence:** 4

**Summary:**

This paper proposes a new regularization technique which applies strong regularization on a dropout-based sub-model instead of the original model. This is because that directly applying the strong regularization on the original model usually leasds to inferior performance. They bypass this by imposing a self-distillation-based relaxed loss to distill knowledge from the original model to the submodel. Experiments mainly on the ImageNet dataset verify the effectiveness of the proposed method.

**Strengths:**

1. This paper is easy to follow. I can understand the basic ideas quickly.
2. The proposed method is simple yet effective. It widely improves the performance of ViTs with a strong training recipe.

**Weaknesses:**

1. The motivation of this paper is clear. It lacks analysis and insight about 1) why imposing strong regularization leads to inferior performance; 2) why using a self-distillation loss and a submodel can solve this problem. Is it because the dropout or the self-distillation loss? What if we use the original loss to train the submodel or additionally use a self-distillation loss to train a strongly regularized model? Basically, it is not clear the relations between regularization、dropuut and self-distillation. No ablation studies about this is provided.
2. The proposed method would increase the training cost by a large proportion.
3. This paper provides experimental analysis with a 100-epoch training recipe. However, from my previous experimental experice, applying regularization on models training with fewer epochs lead to different phenomonons.

**Questions:**

See weakness.

---

### Author Response · Authors · 2023-11-13

Dear Reviewers

Thank you for your insightful and valuable comments.
After careful consideration, we have decided to withdraw our paper.
We plan to revise it based on your feedback for the next submission.
We appreciate all the effort you have put into the reviews.

Best
Authors